# Survival Outcomes of Pancreatic Intraepithelial Neoplasm (PanIN) versus Intraductal Papillary Mucinous Neoplasm (IPMN) Associated Pancreatic Adenocarcinoma

**DOI:** 10.3390/jcm9103102

**Published:** 2020-09-25

**Authors:** Timothy McGinnis, Leonidas E. Bantis, Rashna Madan, Prasad Dandawate, Sean Kumer, Timothy Schmitt, Ravi Kumar Paluri, Anup Kasi

**Affiliations:** 1School of Medicine, University of Kansas Medical Center, Kansas City, KS 66160, USA; t405m343@kumc.edu; 2Department of Biostatistics and Data Science, University of Kansas Medical Center, Kansas City, KS 66160, USA; lbantis@kumc.edu; 3Department of Pathology and Laboratory Medicine, University of Kansas Medical Center, Kansas City, KS 66160, USA; rmadan@kumc.edu; 4Department of Cell Biology, University of Kansas Medical Center, Kansas City, KS 66160, USA; pdandawate@kumc.edu; 5Department of Surgery, University of Kansas Medical Center, Kansas City, KS 66160, USA; skumer@kumc.edu (S.K.); tschmitt@kumc.edu (T.S.); 6School of Medicine, University of Alabama-Birmingham, Birmingham, AL 35233, USA; rpaluri@uabmc.edu; 7Department of Medical Oncology, University of Kansas Medical Center, Kansas City, KS 66160, USA

**Keywords:** pancreatic cancer, PanIN, IPMN, pancreatic adenocarcinoma, precursor lesion

## Abstract

Pancreatic intraepithelial neoplasms (PanINs) and intraductal papillary mucinous neoplasms (IPMNs) are common pancreatic adenocarcinoma precursor lesions. However, data regarding their respective associations with survival rate and prognosis are lacking. We retrospectively evaluated 72 pancreatic adenocarcinoma tumor resection patients at the University of Kansas Hospital between August 2009 and March 2019. Patients were divided into one of two groups, PanIN or IPMN, based on the results of the surgical pathology report. We compared baseline characteristics, overall survival (OS), and progression free survival (PFS) between the two groups, as well as OS and PFS based on local or distant tumor recurrence for both groups combined. 52 patients had PanINs and 20 patients had IPMNs. Patients who had an IPMN precursor lesion had better median PFS and OS when compared to patients with PanIN precursor lesions. However, the location of tumor recurrence (local or distant) did not show a statistically significant difference in OS.

## 1. Introduction

Pancreatic cancer is a leading cause of cancer death in the United States, with an estimated 42,740 deaths in 2019 [1]. Although the survival rate has improved over the last three decades, five-year survival is still only 10% [2]. This poor survival rate is partly due to the fact that 52% of patients have distant metastases at the time of diagnosis. However, if the cancer is still localized at the time of diagnosis (11% of cases) the five-year survival increases to 39.4% [2]. For this reason, early detection of pancreatic cancer is important in improving survival rate. Early identification and treatment of pancreatic cancer and its precursor lesions have been shown to be beneficial in improving survival and cure rates [3]. This is likely due to the fact that early detection of small lesions increases the chance that they will be resectable [4,5,6]. However, early detection of pancreatic cancer, even in high-risk patients, has been difficult [3]. Additionally, even if early detection of small tumors does occur, that does not mean the tumor had not already spread. One study found that 30% of patients with tumors less than 1 cm already had lymph node spread at time of diagnosis [7]. This study also found that 10% of patients with tumors less than 1 cm had distant metastases at the time of diagnosis [7].

Understanding how these small lesions will progress is important, as this can possibly help physicians predict prognosis and overall survival. Therefore, an increased understanding of pancreatic cancer precursor lesions is crucial.

The two main pancreatic cancer precursor lesions are pancreatic intraepithelial neoplasms (PanINs) and intraductal papillary mucinous neoplasms (IPMNs). These lesions have been widely studied and have unique pathological and molecular characteristics [8]. Additionally, IPMNs and PanINs each have a unique pattern of progression into pancreatic adenocarcinoma [9].

Knowledge of the pathology and molecular biology of these specific lesions has been beneficial for physicians because it allows for a treatment plan to be put in place that can be used to prevent progression to pancreatic cancer. Specific guidelines on the management of pancreatic cancer precursor lesions are available [10,11,12].

Although there has been much research into the progression and treatment guidelines of these precursor lesions, there is little information about their specific associations with prognosis and overall survival. Therefore, the goal of this project is to determine if there is an association between the type of precursor lesion and survival rate of pancreatic adenocarcinoma. A better understanding of these associations may have implications in the prognosis, management, and treatment of pancreatic cancer.

## 2. Experimental Section

This study was a retrospective chart review of 72 patients with pancreatic adenocarcinoma who underwent pancreatic tumor resection at the University of Kansas Hospital between August 2009 and March 2019. Only patients who had resectable tumors were included in this study. Resectability of the tumor was determined by the performing surgeon, and no specific criteria were used. Patients were divided into one of two groups: pancreatic adenocarcinoma arising from PanIN or pancreatic adenocarcinoma arising from IPMN. This differentiation was based on the results of the surgical pathology report. Only patients with adenocarcinoma arising from PanIN 2 and PanIN 3 were included in this study. PanIN 2 and PanIN 3 were combined into one group for this study. Baseline characteristics (age, gender, Eastern Cooperative Oncology Group [ECOG] status, tumor location, CA19-9 at time of diagnosis, and Charlson comorbidity index), overall survival (OS), and progression free survival (PFS) were compared between the two groups. Additionally, data on location of tumor recurrence was collected in all patients, when applicable.

Data on pathological stage of the resected tumor was also collected on each patient. Stage was determined using the American Joint Commission on Cancer (AJCC) Staging Criteria 8th Edition. These staging criteria are based on the TNM (T = size and extent of main tumor; N = number of nearby lymph nodes with cancer; M = metastasis) staging of cancer. The pathological stage, along with TNM staging, was always reported in the surgical pathology report. Of note, there were some patient’s that had their tumor resections prior to development of the 8th edition staging criteria. For these patients, we reevaluated the surgical pathology report and updated the pathological stage accordingly. This was done in order for all study subjects to have an equal pathological representation.

In order to address whether there was a need to adjust data for cofounding factors (i.e., age, gender, ECOG status, tumor location, Charlson comorbidity index), the following steps were undertaken. A multivariate analysis was employed to assess whether any of the above covariates affected the estimated survival curves. This analysis was done for both overall survival (OS) and progression free survival (PFS). However, CA19-9 was not included in the model due to the large number of missing values (23.08% for PanIN and 40% for IPMNs) that could compromise the estimators of the remaining coefficients of the variables that are fully recorded.

Exclusion criteria included: (1) any patient with pancreatic adenocarcinoma arising from PanIN 1; (2) any patient with pancreatic adenocarcinoma not arising from IPMN or PanIN 2 or 3; (3) any patient with pancreatic cancer that was not an adenocarcinoma based on the surgical pathology report.

Patients with pancreatic adenocarcinoma arising from PanIN-1 were excluded for two reasons. One reason is that previous studies have found that PanIN 1 lesions have a high occurrence in benign conditions. For example, one study found that 43% of PanIN-1 lesions were found in pancreatic specimens of patients with benign conditions [13]. Additionally, PanIN-1 lesions are ubiquitous and are often incidentally observed in the general population over 50 [8,13,14]. For these reasons, PanIN-1 lesions can often be viewed as indolent. Therefore, there is concern that we cannot confidently say that a certain pancreatic adenocarcinoma arose from a PanIN lesion.

The primary outcomes of interest in this study were overall survival (OS) and progression free survival (PFS) in patients with diagnosed pancreatic adenocarcinoma arising from either PanINs or IPMNs. PFS was calculated from date of pancreatic adenocarcinoma tumor resection to date of progression of the pancreatic cancer after tumor resection. Date of initial biopsy was not used as an indication of date of diagnosis. Progression was determined by evidence of tumor recurrence/relapse/progression on imaging follow-up studies after tumor resection. Overall survival was determined by date of tumor resection to the date of death, or if the patient was not deceased, date of most recent follow-up. In addition to PFS, data on the location of tumor recurrence was collected (i.e., local, distant, or both). The data on location of tumor recurrence was combined for both groups because only two patients with IPMN had local recurrence and both were still alive at date of last follow-up. Therefore, the survival curve could not be defined for each group individually.

Median overall survival and progression free survival between the two groups were compared via a Log-Rank (Mantel-Cox) test to determine if there was statistical significance. An alpha value of 0.05 was considered significant. Additionally, a 95% confidence interval was calculated for OS and PFS in each patient group. OS and PFS are expressed in terms of months. We also analyzed OS and PFS based on local or distant tumor recurrence for both groups combined using the same log-rank method. Survival curves comparing both OS and PFS between both groups were created using the Kaplan-Meier method.

This study was approved by the institutional review board on 28 May 2019. The study was reviewed by the University of Kansas Medical Center Human Research Protection Program. The type of review was a Flexible Institutional Review Board (IRB) review. The study was eligible for Flexible IRB Review because it was deemed minimal risk and is not associated with any federal funding or support. The IRB# is STUDY00144077.

## 3. Results

### 3.1. Characteristics of Patients

A total of 72 patients with resected pancreatic adenocarcinoma between August 2009 to March 2019 were included in this study. Of them, 52 patients had adenocarcinoma arising from PanINs and 20 patients had adenocarcinoma arising from IPMNs (Table 1).

Data on CA (cancer antigen) 19-9 levels at the time of diagnosis were not available for all patients. This was due to one of two reasons. Firstly, this information was not present in the patients’ electronic medical record during data collection. Some patients had outside labs that were not accessible of CA19-9 was never recorded in the EMR. Secondly, some patients never had CA19-9 ordered at the time of diagnosis of their pancreatic cancer. The CA19-9 data in Table 1 were still included for completeness.

Application of a multivariate analysis to determine if there was a need to adjust data for confounding factors (Table 1) found that adjustments for such factors was not necessary. We observed that for all covariates (excluding “group” that refers to PanIN and IPMN) the corresponding coefficients result in *p*-values that are >0.05. The group variable was significant (*p*-values are equal to 0.024 and 0.020 for OS and PFS, respectively). This significance was also assessed by the Kaplan–Meier curves (Figure 1 and Figure 2). Thus, we conclude that the data does not need adjustments for confounding factors.

### 3.2. Survival Outcomes

The median overall survival for patients with pancreatic adenocarcinoma arising from PanINs was 70.3 months (95% CI: 35.4–105.2), and for tumors arising from IPMNs median OS was 78.8 months (95% CI: 33.2–124.4). The difference in OS between these two groups was found to be statistically significant with a *p*-value of 0.013 (Figure 1).

The median progression free survival for patients with tumors arising from PanINs was 26.2 months (95% CI: 21.4–31.0), and for tumors arising from IPMNs median PFS was found to be 74.3 months (95% CI: 15.7–132.9). The difference in PFS between PanIN and IPMN was also found to be statistically significant with a *p*-value of 0.004 (Figure 2).

Overall, 33 patients (46%) were found to have recurrence of pancreatic cancer in this study. Of those, 11 had local recurrence and 22 had distant recurrence. Local recurrence in the PanIN group occurred in nine patients and distant recurrence occurred in 17 patients. Local recurrence in the IPMN group occurred in two patients and distant recurrence occurred in five patients. The location of tumor recurrence (local versus distant) for both IPMNs and PanINs combined did not show a statistically significant difference in OS (*p*-value 0.33).

Within the PanIN group, median OS after recurrence was 71.3 months (95% CI: 68.8–73.4) for local recurrence and 46.7 months (95% CI: 39.2–54.2) for distant recurrence (*p* = 0.330). Location of tumor recurrence for IPMNs and PanINs combined was also determined for PFS, but there was no statistical difference found (*p*-value 0.064).

Additionally, in order to address the impact that peri-operative mortality may have had on PFS and OS, we examined which patients, if any, died in the peri-operative period (defined as death within 30 days of tumor resection). Peri-operative death occurred in only one patient (due to post-operative complications). In order to determine if this study subject impacted our data, we removed them from the study population and re-analyzed the data using our original log-rank (Mantel-Cox) test. We found that the log rank test for PFS yielded a *p*-value of 0.004 which implied a statistically significant difference between the two groups. For OS, the log rank test yielded a *p*-value of 0.015 which implied a statistically significant different between the two groups.

We observe that this person is censored with respect to progression and is not censored with respect to overall survival. As such, it does not change the results of the recurrence survival analysis and slightly changes the overall survival results only for the PanIN group where the individual belongs. In particular, we see that the mean survival changed from 59.630 to 60.860 which is negligible. The median estimate remains unchanged (due to the stepwise nature of the Kaplan-Meier survival estimate). Thus, the exclusion of the individual does not affect the analysis.

## 4. Discussion

The goal of this study was to determine if there is an association between type of pancreatic adenocarcinoma precursor lesion (IPMN versus PanIN) and overall survival and progression free survival. Results from this study show that patients with pancreatic adenocarcinoma arising from IPMNs have statistically significant longer OS and PFS when compared to patients with tumors arising from PanINs. This finding is significant, as it can provide physicians with a better understanding of patient prognosis after tumor resection.

A recent systematic review of pancreatic cancer resection data found that median overall survival for stage 1A (localized, resectable) pancreatic cancer was between 24 and 42 months [15]. However, this study did not differentiate between pancreatic cancer arising from PanIN or IPMNs. Therefore, data from our study can provide physicians with more specific prognosis indication. Additionally, this data may eventually influence adjuvant treatment options after tumor resection. However, further studies regarding specific treatment for pancreatic cancer arising PanIN versus IPMN need to be done. Additionally, advancement in imaging technology and risk factor assessments have improved early detection of precursor lesions [16]. Therefore, knowledge of how these lesions will progress may improve patient treatment options and outcomes.

The progression to pancreatic adenocarcinoma from precursor lesions has been well studied [17,18,19,20]. Differences in progression of pancreatic cancer arising from IPMN versus PanIN may account for the differences in overall survival and progressive free survival between the two groups. For example, one study found that patients with IPMNs have a lower prevalence of TP53 mutations [21]. TP53 mutations were present 9.1% of IPMNs, 17.8% in PanIN 2, and 38.1% in PanIN 3 [21]. Mutations in TP53 have been found to confer a worse prognosis in patients with pancreatic cancer. This study demonstrated that patients with lower levels of TP53 mRNA expression had a statistically significant worse prognosis [22]. This may partly explain why patients with adenocarcinoma arising from PanINs have lower median OS and PFS.

Another possible explanation for the increased OS in the IPMN patients is that not all IPMNs behave the same. Previous research has shown that IPMNs that arise from branches of the main pancreatic duct seem to be more indolent compared to lesions that arise in the main duct [23,24]. It is possible that many of the IPMNs in our study may have arisen from the branch ducts, which could account for an improvement in OS. However, the pathology reports used to collect data did not always specify where the IPMN arose from, so data on this point was not collected. Therefore, this is nothing more than a possible theory as to why IPMN patients had better OS than PanIN patients. Additional studies with larger IPMN populations could investigate this theory by stratifying the IPMN patient group into main duct and branch duct and then comparing their respective OS and PFS to a PanIN patient group.

The results of our study, while significant, still need further validation with additional studies and/or in a larger patient population. However, our findings do support a previous report that examined survival post-resection in patients with pancreatic adenocarcinoma arising from PanIN versus IPMN [25]. This report found that five-year survival post-resection in PanIN patients was 15–20% and 10-year survival was <2%. Five-year survival post-resection in the IPMN patients was significantly higher at about 60%, and 10-year survival was around 40%. This study did stratify the location of the IPMNs into main duct and branch duct, but five- and 10-year survival was similar between the two groups.

An additional question that may warrant further exploration is whether adjuvant chemotherapy and/or radiation post-tumor resection has any impact of OS and PFS. This would likely need to done with a large patient population in order to ensure statistically significant data. Certain chemotherapy regimens also could be analyzed to determine if they have an impact on OS and PFS in PanINs versus IMPNs. However, this is beyond the scope of this study.

The limitations of this study include that it is a retrospective chart review with a relatively small patient population (*n* = 72). Therefore, the common limitations of retrospective cohort studies apply to this study. These include the absence of data from the electronic medical record (i.e., CA19-9 was not reported in all patient charts) and the inability to control treatment regimens for each patient. This study was relatively small with 72 subjects. Although the main results were still found to be statistically significant, further studies with larger patient populations will be needed to confirm our findings. Furthermore, this study is only a one-institution experience, and other institutions may have different patient populations or different outcomes.

Additionally, no data were collected on prior surveillance of patients with pancreatic adenocarcinoma arising from IPMN. This may be viewed as a limitation, as prior surveillance of IPMNs could impact overall survival. If IPMN patients were followed prior to diagnosis of pancreatic adenocarcinoma, it likely would mean their disease was caught at an early stage. Therefore, this could improve survival. This is a possible confounding factor that could be further investigated in future studies, possibly with a sub-analysis of IPMN patients with prior surveillance.

## 5. Conclusions

The results of this study are encouraging in that they provide important information regarding prognosis and overall survival in patients with pancreatic cancer. Pancreatic adenocarcinoma carries a dismal prognosis. However, increasing our understanding of the precursor lesions may help to improve treatment options and survival outcomes.

## Figures and Tables

**Figure 1 jcm-09-03102-f001:**
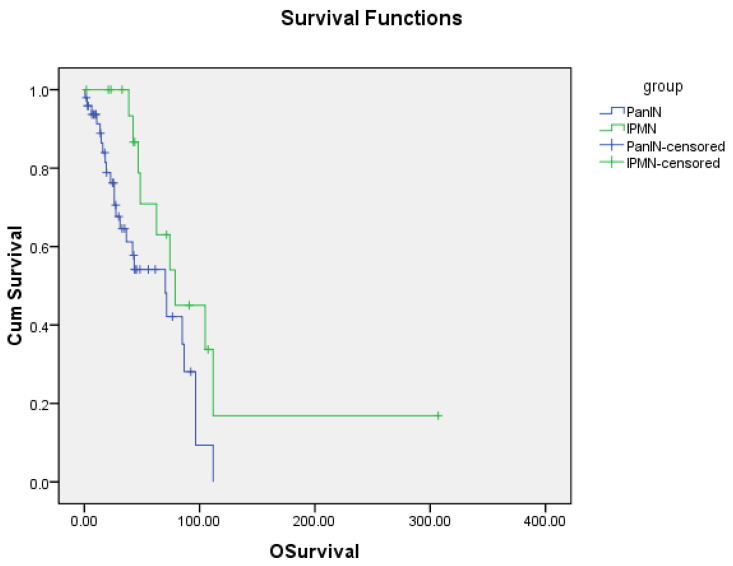
Overall survival of PanIN and IPMN. Osurvival—overall survival; Cum Survival—cumulative survival; PanIN—Pancreatic Intraepithelial Neoplasia; IPMN—Intraductal Papillary Mucinous Neoplasia.

**Figure 2 jcm-09-03102-f002:**
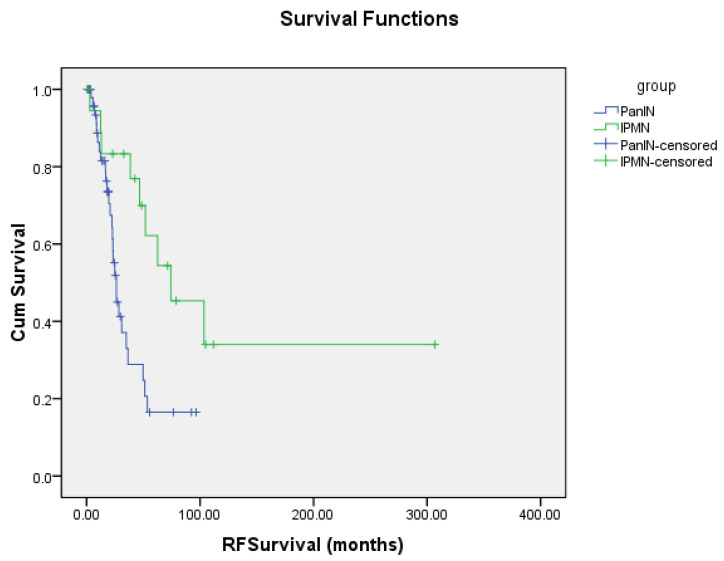
Progression free survival of PanIN and IPMN. RFSurvival—Progression Free Survival

**Table 1 jcm-09-03102-t001:** Baseline Characteristics.

Characteristics	PanIN (*n* = 52)	IPMN (*n* = 20)
Median Age (years)	62.5 (42–85)	69 (54–77)
Gender (%)		
Female	19 (37%)	7 (35%)
Male	33 (63%)	13 (65%)
ECOG Status		
0–1	51 (98%)	17 (85%)
2 or higher	1 (2%)	3 (15%)
Tumor Location		
Head	45 (87%)	14 (70%)
Body	3 (6%)	3 (15%)
Tail	4 (7%)	3 (15%)
CA19-9 at time of diagnosis		
Normal (<37)	7 (21%)	4 (33%)
Abnormal	26 (79%)	8 (67%)
Charlson Comorbidity Index (median)	5	5
Pathological Stage (AJCC Staging Criteria)		
Stage IA	7	4
Stage IIA	2	2
Stage IB	5	9
Stage IIB	22	3
Stage III	16	2

PanIN—Pancreatic Intraepithelial Neoplasia; IPMN—Intraductal Papillary Mucinous Neoplasia; ECOG—Eastern Cooperative Oncology Group; CA—Cancer antigen; AJCC—American Joint Committee on Cancer.

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
