# Peer review of "Survival Outcomes of Pancreatic Intraepithelial Neoplasm (PanIN) versus Intraductal Papillary Mucinous Neoplasm (IPMN) Associated Pancreatic Adenocarcinoma"

_jcm, 2020, doi:10.3390/jcm9103102_

Round 1

Reviewer 1 Report

McGinnis et al. investigated a total of 72 patients with resected pancreatic adenocarcinoma between August 2009 to March 2019. 52 patients had adenocarcinoma arising from PanINs and 20 patients had adenocarcinoma arising from IPMNs.

Authors concluded that patients who had a IPMN precursor lesion had better median PFS and OS when compared to patients with PanIN precursor lesions. However, the location of tumor recurrence (local or distant) did not show statistically significant difference in OS.

Despite its retrospective nature and the limited sample size, this is a very well written manuscript on a very original topic.

Probably, a propensity-score-matched analysis would reduce the effect of possible confounding factors and bias among the two different groups.

Reviewer 2 Report

In their manuscript entitled "Survival outcomes of pancreatic intraepithelial neoplasm (PanIN) versus intraductal papillary mucinous neoplasm (IPMN) associated pancreatic adenocarcinoma" the authors present a retrospective institutional experience of a cohort of 72 patients operated for PanIN (2&3) and IPMN patients with the aim to determine if there is an association between the type of precursor lesion and survival rate of PDAC. Based on their outcomes the authors concluded that patients who had a IPMN precursor lesion had a better median PFS and OS when compared to patients with PanIN precursor lesions and that location of tumor recurrence did not show statistically significant difference in OS.

This is a somewhat interesting study, based on a small number of patients, which however has some significant problems including:

  • The baseline characteristics as well as disease staging outcomes are not adequately given to the readership ie. what about T or N status?
  • What about perioperative mortality?
  • What about adjuvant therapy?
  • Were any of the IPMN patients under surveillance previously?
  • The limitations of this study are very poorly described
  • IRB approval?
  • There are missing data for CA19-9
  • Why did the authors choose to report CCI for these patients?
  • The discussion is too short

Reviewer 3 Report

This is a well-written manuscript that compares survival outcomes between patients with pancreatic adenocarcinomas that arose from either PanIN or IPMN at a single center. The manuscript is concise and the data are appropriately presented. I believe there is a significant flaw with respect to the patient inclusion criteria that should be corrected prior.

The authors exclude patients who were found to have pancreatic adenocarcinoma "arising from PanIN 1" lesions. I do not believe that it is appropriate to exclude patients with PanIN 1 lesions. Pancreatic intraepithelial neoplasia is described as a spectrum of lesions progressing from PanIN 1 to PanIN 3 and on to pancreatic adenocarcinoma. As a lesion advances through this spectrum it accumulates more histologic abnormality and an increased number of somatic genetic aberrations. Each step in this progression is associated with specific histologic and genetic findings. Since all the patients selected for this study have a pathologic diagnosis of pancreatic ductal adenocarcinoma, they have progressed through this full spectrum. The PanIN lesions found in the pathologic resection specimens are used to determine whether the carcinoma came from an PanIN precursor or an IPMN precursor, but the patients have already progressed through this spectrum. Therefore, it makes no sense to exclude patients with PanIN 1 lesions.

The analysis needs to be repeated including patients with PanIN 1-3 detected in addition to their adenocarcinomas.

Round 2

Reviewer 2 Report

The authors presented with a massively improved revised manuscript, answering most of my questions. Two points to correct:

  • CCI is Charlson Comorbidity Index and NOT Charleston comorbidity index
  • The reason I asked if IPMN patients were under surveillance previously is quite obvious. Although the authors mentioned this limitation I do not believe they make it clear to the readership. A patient who is followed up for a period of time is more likely to be diagnosed with early stage disease and therefore have improved survival. Therefore "Therefore, any prior knowledge of an IPMN precursor lesion would not likely have directly impacted our definition of OS" is incorrect in my view.

Author Response

Point 1: CCI is Charlson Comorbidity Index and NOT Charleston comorbidity index

Response 1: This will be corrected in the manuscript. Thank you for pointing this out.

Point 2: The reason I asked if IPMN patients were under surveillance previously is quite obvious. Although the authors mentioned this limitation I do not believe they make it clear to the readership. A patient who is followed up for a period of time is more likely to be diagnosed with early stage disease and therefore have improved survival. Therefore "Therefore, any prior knowledge of an IPMN precursor lesion would not likely have directly impacted our definition of OS" is incorrect in my view.

Response 2: You are correct. We did not clearly address this limitation in the revised manuscript, and we will ensure to clarify in the next revision. We will also discuss how this could be addressed in future studies. These revisions appear in lines 259-265 in the revised manuscript. We hope we adequately addressed this concern. Thank you for the input.

Reviewer 3 Report

All my concerns and the concerns of the other reviewers have been sufficiently addressed.

Author Response

Thank you.